# Shoot Nutrition and Flavor Variation in Two *Phyllostachys* Species: Does the Quality of Edible Bamboo Shoot Diaphragm and Flesh Differ?

**DOI:** 10.3390/foods12061180

**Published:** 2023-03-10

**Authors:** Lili Fan, Junjing Hu, Ziwu Guo, Shuanglin Chen, Qijiang He

**Affiliations:** 1Research Institute of Subtropical Forestry, Chinese Academy of Forestry, Hangzhou 311400, China; 2Hangzhou Academy of Forestry, Hangzhou 310005, China

**Keywords:** bamboo shoot components, bamboo species differences, bamboo shoot nutrients, bamboo shoot palatability, amino acid flavor compounds

## Abstract

For their quality evaluation, it is essential to determine both bamboo shoot nutrition and palatability, which will have a decisive effect on their economic value and market potential. However, differences in shoot nutrition and flavor variation among bamboo species, positions, and components have not been scientifically validated. This study assessed nutritional and flavor differences in two components (i.e., shoot flesh (BSF) and diaphragm (BSD)) of two *Phyllostachys* species (i.e., *Phyllostachys edulis* and *Phyllostachys violascens*) and analyzed any positional variation. Results showed that BSF protein, starch, fat, and vitamin C contents were comparatively higher. Nutrient compounds in the upper shoot segment of *Ph. edulis* were higher and contained less cellulose and lignin. However, both species’ BSD total acid, oxalic acid, and tannin contents were comparable. BSD soluble sugar and sugar:acid ratio were higher than upper BSD total amino acid, four key amino acids (i.e., essential amino acid, bitter amino acid, umami amino acid, and sweet amino acid flavor compounds), and associated ratios were all higher than BSF while also being rich in amino acids. The content and proportion of BSF essential and bitter amino acid flavor compounds in *Ph. edulis* were high relative to *Ph. violascens*. Conversely, the content and proportion of BSD umami and sweet amino acid flavor compounds were comparable to that of *Ph. edulis*. Our results showed that bamboo shoot quality was affected by flavor compound differences and that interspecific and shoot components interact. This study offers a new perspective to determine the formative mechanisms involved in bamboo shoot quality while providing a basis for their different usages.

## 1. Introduction

Bamboo shoots, being high in fiber and low in fat, rich in amino acids, and an available green and healthy food product, are deeply entrenched in people’s culinary ethos, particularly in Asia [1]. Bamboo shoots are a traditional Chinese forest vegetable and a primary agricultural product exported worldwide. Moreover, the high-quality vitamins, carbohydrates, proteins, and minerals contained in bamboo shoots and their general ease of accessibility can help resolve nutritional deficiencies endemic in poor rural communities [2]. However, with people’s ongoing pursuit of happiness and health, the demand and consumption of high-quality bamboo shoots have exploded in popularity. The nutritional benefits and palatability (i.e., flavor) of bamboo shoots have consequently, become essential factors that impact market competitiveness while acting to restrict their economic value [2,3]. Ultimately, improving and maintaining bamboo shoots’ nutritional quality and palatability are crucial objectives in bamboo management and a meaningful way to improve the economic benefits of bamboo forests.

Bamboo species are characterized by their high growth rate after their shoots first emerge from the soil as well as their singular culm growth habit [4]. The culm of most bamboo species has extraordinarily long and hollow stalks (i.e., internodes), which are segmented into many nodes, being densely distributed at the root and top segments while loosely distributed at the middle segment [5]. The bamboo node diaphragm structure is an adaptive means by which bamboo has reinforced an essential biological feature [5,6,7]. For its survival, it has been suggested that the persistent distribution of the bamboo diaphragm structure functions as a critical lateral transport channel for water and nutrient uptake while enhancing mechanical longitudinal internode growth performance, improving the stiffness and strength of the culm as a whole [8,9,10]. However, previous studies have only described the anatomical structure of the diaphragm and associated physical and physiological properties [11,12,13]. Thus, there is currently limited information on the status of bamboo shoot diaphragm nutrition and palatability.

After the removal of the diaphragm, shoot flesh is the main edible part of the bamboo shoot. Generally, studies on bamboo shoot quality have rarely distinguished between flesh and diaphragm components, hindering directional utilization and mining targeted functional substances. Moreover, overt anatomical differences between the bamboo diaphragm and wall result in noticeable functional differences [9,14]. For example, the vascular bundle duct at the bamboo diaphragm is repeatedly twisted and horizontally rotated, forming a complex network structure that protects bamboo flesh from mechanical stress and enables the lateral transportation of water and nutrients [8,9,11]. Bamboo shoot flesh derives from the distribution area of the intercalary meristem. Moreover, the degree of differentiation between bamboo flesh ductal cells is higher than other bamboo cells. Cell vacuolization is also higher, and bamboo maturity is related to the longitudinal transport of critical nutrients and water during periods of high shoot growth [8,15,16]. Structural and functional changes in bamboo shoot diaphragm and flesh will impact the fundamental mechanics of material metabolic and physiological processes during shoot development, leading to changes in shoot quality. However, shoot diaphragm and flesh quality discrepancies have not been scientifically validated.

Determining bamboo shoot quality necessitates comprehensively evaluating its many components (e.g., protein, sugar, oxalic acid, phenols, fibers, amino acids, etc.) [17,18,19,20]. Thus far, differences in nutritional composition and flavor compounds that underlie bamboo’s shoot diaphragm and flesh remain largely unexplored. Accordingly, this study investigated the nutritional and flavor quality status of two key structural bamboo components (i.e., shoot diaphragm and flesh) of two *Phyllostachys* species (i.e., *Phyllostachys edulis* and *Phyllostachys violascens*), where both monopodial bamboo species are characterized by their high yields and economic value. This study aimed to clarify structural component effects on bamboo shoot quality. Here, we identified the biological characteristics of bamboo shoots related to component changes, including nutritional features, flavor compounds, and the content of various amino acids, to answer the following questions: (1) Is there an appreciable component effect on *Phyllostachys* shoot quality (i.e., nutrition and palatability) following shoot diaphragm and flesh separation? (2) Do interspecific differences and positional changes cause corresponding quality differences in the diaphragm and flesh of these two bamboo species? This study intends to help to better understand the developmental mechanisms associated with bamboo shoot quality while providing a conceptual reference for the directional cultivation and the scientific utilization of high-quality bamboo shoots.

## 2. Materials and Methods

### 2.1. Study Site

The Taihuyuan Ornamental Bamboo Garden was selected as the study area. It is in Lin’an City, Zhejiang Province, China, in the northwestern region of Zhejiang Province, east of Hangzhou City and west of Huangshan (30°18′ N lat and 119°37′ E long) (Figure 1). The area is influenced by a warm and humid monsoon climate. It experiences sufficient sunlight, with an average annual temperature of 16.5 °C, an extreme maximum temperature of 37.9 °C, and an extreme minimum temperature of −5.7 °C. The yearly sunshine hours are 1437 h, annual precipitation is 1613 mm, and the yearly frost-free period is 237 days. Red loam is the primary soil type, with a soil pH between 5.0–6.5. Fertilizer retention performance is good. The Taihuyuan Ornamental Bamboo Garden was established in 2002. Its total area is >70 ha, comprised of >250 bamboo species types.

### 2.2. Sampling Material Description

*Ph. edulis* and *Ph. violascens*, the most important monopodial bamboos, are both the most famous shoot-used species with a total area of 4.5 million hectares in China, with obvious shoot quality differences [21,22]. For this study, four 10 m × 10 m plots were established within the *Ph. edulis* and *Ph. violascens* distribution area of the Taihuyuan Ornamental Bamboo Garden. We randomly excavated ten bamboo shoots from each plot (10–15 cm in length). Soil and debris were removed from shoot samples. Following this, sheaths were peeled from the shoots before they were washed in ultrapure water (UPW). Finally, the inedible base of the shoots was removed. The fresh bamboo shoot samples were divided into three segments (i.e., upper, middle, and base), where the bamboo shoot diaphragm (BSD) and flesh (BSF) were separated at each position. BSD was taken from the inner and transverse solid septum of the shoot node area, and BSF was the outer part of the shoot body, including the tip, inter-nodal, and nodal components, after removing BSD. Each replicate was derived from mixing ten bamboo shoot samples, providing four replicates. Finally, nutritional and taste quality indices were measured (Figure 1).

Fresh bamboo shoot flesh (BSF) and diaphragm (BSD) samples from each bamboo segment were ground into a homogenate in a component grinder to determine protein, oxalic acid, total acid, tannins, vitamin C, and free amino acid contents. Another fresh sample section was oven-dried to constant weight at 60 °C before being ground into powder and passed through a 0.4-mm sieve to determine fat, soluble sugar, starch, cellulose, and lignin contents (*n* = four replicates).

### 2.3. Determination of Nutritional Indicators

Protein content was evaluated as described by the method of Xu et al. [22]. The homogenized suspension (5.0 g) was digested by adding 0.4 g copper sulfate, 6.0 g potassium sulfate, and 20 mL sulfuric acid. When the temperature of the digester reached 420 °C, the suspension was allowed to continue digestion for 1 h. The liquid was cooled when green and clear, and 50 mL of water was added. The protein content was measured using a Kjeldahl nitrogen analyzer (SKD-2000, Shanghai Peo Analytical Instruments Co., Ltd., Shanghai, China).

Fat content was determined through Soxhlet extraction [23]. The sample was placed in a filter paper tube with 2 g of ground powder, and the tube was tied firmly with cotton thread and placed into the extraction tube of the Soxhlet extraction apparatus. At the extraction end, one drop of extraction solution was picked up with a frosted glass piece, and the absence of oil spots on the frosted glass piece indicated that the extraction was complete and the solvent was recovered. One to two mL of solvent remaining in the receiving flask was evaporated in a water bath, dried at 103 °C, cooled in a desiccator for one hour, and weighed. This step was repeated until a constant weight was achieved (the difference between the two weights did not exceed 2 mg). The fat content was expressed in g·100 g^−1^ dry weight.

The vitamin C content was determined through UV spectrophotometry [24]. Five grams of sample homogenate was added to a 2% acetic acid solution (1:2) and extracted for 30 min in a shaded environment. The extract was centrifuged (4000 r·min^−1^) for 10 min, and the supernatant was adjusted to 100 mL. The absorbance at the maximum absorption wavelength was measured at 243 nm.

The starch content was observed via anthrone colorimetry using the method of Gao et al. [25] with minor changes. A 0.5 g dry sample was added to sufficient water and maintained in a boiling water bath for 5 min. Then, to the mixed solution was added a sufficient amount of sodium hydroxide solution to make the mixture appear dark blue, and the extract solution was diluted by distilled water ten times. The absorbance of starch was measured by a UV-visible photometer (TU-1901, Beijing Puxi General Instrument Co., Ltd., Beijing, China) at 620 nm.

The cellulose and lignin contents were measured by a kit (Suzhou Keming Biotechnology Co., Ltd., Beijing, China). A total of 0.01 g and 0.005 g dried samples were weighed into a 1.5 mL EP tube and 10 mL glass test tube to extract cellulose and lignin into solution, which were then measured by absorbance at 620 nm and 280 nm, respectively. The cellulose and lignin contents were expressed as mg·g^−1^ dried weight.

### 2.4. Determination of Taste Quality Indicators

Oxalic acid content was determined through reversed-phase high-performance liquid chromatography [26]. A total of 0.5 g fresh sample was added to 2 mL of 0.5 mol·L^−1^ HCl and a small amount of quartz sand, to enable thorough grinding into a homogenate, which was then poured into a test tube and boiled in a water bath for 15–20 min, shaken, cooled, and added to 4 mL of distilled water and left overnight. The homogenate was filtered into a 50 mL volumetric flask with a small funnel. The residue was repeatedly washed with distilled water to 50 mL. Finally, 1 mL of solution was analyzed using a High-Performance Liquid Chromatograph (Shimadzu LCMS-2010, Shimadzu (Shanghai) Global Laboratory Consumables Co., Ltd., Shanghai, China) after filtering through a microporous membrane (aqueous system, Φ 0.45 m).

Total acid content was determined by the titration method [22]. A total of 200 g fresh sample was placed in a 500 mL beaker and shaken for 3–4 min under reduced pressure to remove CO_2_. Sample solution (25.0 mL) was pipetted into a 250 mL volumetric flask, distilled water added, placed on the scales, shaken well, and filtered. A total of 25 mL of the filtered solution was placed in a 250 mL triangular flask, two to four drops of phenolphthalein indicator solution added, and titrated with 0.1 mol·L^−1^ sodium hydroxide titration solution until it was slightly red and color maintained for 30 s without fading, and the volume of sodium hydroxide titration solution was recorded for calculating the total acid content.

Tannin content was determined by the colorimetry method [27]. A 0.05 g sample with 80 mL of distilled water was boiled in a water bath for one hour. After cooling, the solution was adjusted to 100 mL with distilled water and centrifuged for 4 min (8000× *g*). A 1 mL sample of the supernatant was mixed with 5 mL of distilled water, 1 mL of a mixture of sodium tungstate and sodium molybdate, and 3 mL of sodium carbonate solution (75 g·L^−1^), and the absorbance was measured at 765 nm.

Soluble sugar content was also measured using the anthrone colorimetric method [25], and extracted by the same method as starch. The absorbance of soluble sugar extract was measured at 660 nm. Additionally, the sugar:acid ratio represented the soluble sugar to total acid ratio.

### 2.5. Determination of Free Amino Acids Contents

The free amino acids were measured as described by Guo et al. [28]. To a 0.10 g sample, was added 10 mL of 6 mol·L^−1^ HCl solution, hydrolyzed in an oven at 110 °C for 24 h, cooled, and the volume adjusted to 50 mL with pure water. The solution was filtered with a 0.45 μm membrane, 200 μL of the filtered liquid was placed in a large-mouth centrifuge tube, put in the oven again at 60 °C and concentrated until there was no liquid, then 1 mL of 0.02 mol·L^−1^ HCl was added and mixed to obtain the sample solution, which was measured and analyzed using an amino acid analyzer (HITACHI L-8900, Japan).

According to the content of the various amino acid compounds used for our calculations, essential amino acid compounds included valine (Val), methionine (Met), L-isoleucine (Ile), leucine (Leu), and tyrosine (Tyr); bitter amino acid compounds included Ile, Leu, Tyr, phenylalanine (Phe), and Val; umami amino acid compounds included aspartic acid (Asp) and glutamic acid (Glu); sweet amino acid compounds included threonine (Thr), serine (Ser), glycine (Gly), alanine (Ala), and proline (Pro) [28]. The ratio of essential amino acids, bitter amino acids, umami amino acids, and sweet amino acids flavor compounds was calculated based on the proportion of each amino acid to the total amino acid content.

### 2.6. Data Analysis

All data were statistically analyzed in SPSS Statistics 22.0 using One-Way ANOVA, followed by multiple comparison tests (i.e., the least significant difference (LSD) and Tukey’s multiple comparison tests) to determine any significant differences (α = 0.05). Multivariate analysis of variance (MANOVA) was used to test the main and interactive effects of each indicator under species (two levels), components (two levels), and positions (three levels). Prism 8.0.1 and Microsoft Excel 2016 were used to visualize and generate tables, respectively.

## 3. Results

### 3.1. Shoot Diaphragm (BSD) and Flesh (BSF) Nutritional Status of the Two Bamboo Species

The shoot diaphragm and flesh nutritional status comparison between these two bamboo species are summarized in Figure 2A–F. Results showed that both species’ protein, starch, fat, and vitamin C content were significantly responsive to changes in components, positions, and associated interactions (where *p* < 0.05 for all indicators, Figure 2A–D). The BSF protein and vitamin C content of both bamboo species were higher than that of the BSD content, which significantly decreased directionally from the upper to base segments (*p* < 0.001). Generally, the protein and vitamin C content of *Ph. violascens* was higher than that of *Ph. edulis*, while, compared to BSD, the BSF of both species was richer in starch and fat content. However, the starch and fat content in the upper shoot segment of *Ph. edulis* was significantly higher compared to all other segments or positions (*p* < 0.05). Compared with *Ph. edulis*, there was a decrease in starch and fat content in the BSD and the upper BSF segment of *Ph. violascens*. Additionally, no visible component, species, positional, or interactive cellulose response was observed. However, lignin closely correlated to species and position (where *p* < 0.05 for all indicators, Figure 2E,F) but did not respond to associative interactions. Contrary to other indicators, both cellulose and lignin content was lowest in the upper BSD and BSF segments of both species. Compared to *Ph. edulis*, the lignin content in the different shoot segments of *Ph. violascens* decreased, opposite to cellulose.

### 3.2. Shoot Diaphragm (BSD) and Flesh (BSF) Taste Quality of the Two Bamboo Species

After exploring bamboo shoot component effects on nutritional quality, we analyzed taste quality to determine the two bamboo species’ shoot flavor and palatability status (Figure 3A–E). Total acid content significantly correlated with the bamboo species (*p* < 0.05) and the shoot position (*p* < 0.001) while unresponsive to shoot components (Figure 3A). Comparatively, shoot components had a significant effect on oxalic acid and tannin content (*p* < 0.001) while exhibiting strong species and positional interactive effects (*p* < 0.001) (Figure 3B,C). At the same component shoot position, both species’ BSD total acid, oxalic acid, and tannin content were significantly lower than that of BSF (*p* < 0.05). For BSF, total acid and oxalic acid content decreased directionally from both bamboo species’ upper to base segments. In contrast, tannin content was significantly higher in the middle shoot segment compared to the upper (26.87%) and base (13.11%) segments of *Ph. edulis* (*p* < 0.05). However, for BSD these three indicators exhibited no significant positional differences between the bamboo species. Moreover, oxalic acid and tannin content in *Ph. violascens* shoots were lower than that of *Ph. edulis*, differing from total acid content.

As shown in Figure 3D,E, although we observed no significant main component effects for soluble sugar and sugar:acid ratios, they positively correlated with species (*p* < 0.05) and position (*p* < 0.001). Contrary to the base segment, BSD soluble sugar and the sugar:acid ratio in the upper and middle shoot segments were higher than that of BSF in both bamboo species. Additionally, soluble sugar and the sugar:acid ratio in BSD and BSF significantly increased as shoot height decreased (*p* < 0.05). We also observed a significant increase in soluble sugar content and the sugar:acid ratio in BSD (by 44.16% and 38.35%, respectively) and BSF (by 48.60% and 50.89%, respectively) in the shoot base segment of *Ph. edulis* (i.e., relative to *Ph. violascens*), which was opposite to that of the upper segment.

### 3.3. Amino Acid Composition and the BSD and BSF Ratios of the Two Bamboo Species

In addition to determining nutritional and taste quality, we investigated the effects of different bamboo shoot components on amino acid composition and associated ratios (Figure 4A–E and Figure 5A–D). The total amino acid content and composition of the four key amino acids (i.e., essential amino acid, bitter amino acid, umami amino acid, and sweet amino acid flavor compounds) responded significantly to changes in species (*p* < 0.05), component (*p* < 0.001), position (*p* < 0.001), and associated interactive effects of bamboo species and components (*p* < 0.001) and components and position (*p* < 0.05) (Figure 4). BSD total amino acid content and the composition of the four key amino acids in both bamboo species were lower than that of BSF. Additionally, BSD and BSF total amino acid content and the composition of the four key amino acids in both bamboo species increased significantly from the base to the upper segments (*p* < 0.05). *Ph. edulis* shoot components were higher in total amino acid, essential amino acid, and bitter amino acid flavor compounds compared to *Ph. violascens*. Conversely, BSD sweet amino acid compounds and the subsequent flavor of *Ph. violascens* were higher than that of *Ph. edulis*.

The main and interactive effects of bamboo species, shoot components, and shoot positions were highly significant for the proportion of all four amino acids (where *p* < 0.001 for all indicators, Figure 5). For the same shoot component, compositional percentages of the four amino acids in the BSD of both bamboo species were lower compared to BSF. Additionally, BSD essential and bitter amino acid ratios in the middle segment of both bamboo species were comparatively higher than in the upper and base segments. On the other hand, BSF essential and bitter amino acid ratios decreased as shoot height increased. However, proportionally, the trend in BSD and BSF components and sweet amino acid flavor compounds was the opposite of that of essential and bitter amino acid ratios. Moreover, *Ph. edulis* shoot essential and bitter amino acid ratios were higher than the corresponding *Ph. violascens* ratios while proportionally opposite umami and sweet amino acid ratios.

## 4. Discussion

Bamboo shoot nutrition and palatability reflect their economic value and market potential and are key indicators to evaluate their quality [2,3]. Bamboo shoots are mainly composed of BSF and BSD, with obvious structural and functional differences that could be associated with their different material composition [8,15,16]. In particular, the lamellar shoot diaphragm, a unique feature of the node, plays a biomechanical role in the longitudinal growth of bamboo shoots while also controlling horizontal transport (i.e., nutrients and water) [5,7,9,10]. Thus far, little attention has been paid to nutrient and palatability differences between bamboo shoot diaphragm and flesh (i.e., BSD and BSF, respectively) components. Studies have largely ignored the role that BSD plays during shoot quality formation. Accordingly, our study provides a new perspective on the internal and external components of the quality and palatability of bamboo shoots.

In this study, the two bamboo species’ BSF and BSD nutrition status significantly differed, exhibiting positive positional effects. Generally, the BSF protein, starch, fat, and vitamin C content of both bamboo species was higher than that of BSD, indicating that the nutrient accumulation capacity of the former was considerably higher. This may be because BSF is the main distribution area of intermediate meristems, and a well-developed vascular tube that can accommodate higher water conductivity and water content can meet the longitudinal transport nutrient requirements during bamboo shoot growth [12,29,30]. However, the different BSF segments of *Ph. violascens* were high in protein and vitamin C and low in fat, whose nutritional quality exceeded that of *Ph. edulis*, which indicates the obvious genetic differences with respect to internal bamboo shoot component quality maintenance, even within the same genus. The growth of apical bamboo shoot components occurs earlier and embodies strong apical dominance, inhibiting nutrient accumulation in the base segment [31]. BSF and BSD nutritional levels in the upper shoot segment were higher than the middle and base segments in mature bamboo shoots, corresponding to the nutrient-inhibition hypothesis [32,33].

Cellulose and lignin accumulation gradually decreased directionally with bamboo shoot growth, demonstrating that bamboo shoot roughness steadily reduced. Previous studies have reported on lignocellulose allocation in bamboo shoots [34,35]. These studies observed a progressive increase in the maturational status of tissues (from the tip to the base) of *Ph. edulis* and *Asparagus officinalis*, respectively. Here, we also observed a relative cellulose increase and lignin decrease in *Ph. violascens* (i.e., compared to *Ph. edulis*). This indicated that softening and immature phenomenon would likely occur in *Ph. violascens* shoot tissues under the same developmental height and hence be more suitable as a fresh food product. Other studies have observed similar results. For example, studies have found that the cell flesh of developing tissue contained more pectic polysaccharides and major hemicelluloses than that found in the mature tissue of *Brassica oleracea* [36] and *Ph. edulis* shoots [34], respectively. Therefore, compared to *Ph. violascens*, the lignin content in *Ph. edulis* shoots was higher, and this is due to either genetic differences or material morphological requirements [37], making it more suitable for use in future industrial applications [38]. Interestingly, the higher cellulose and relatively lower lignin content of *Ph. violascens* caused its BSD to taste more delicate and tender relative to BSF, which was the opposite outcome for *Ph. edulis*. We speculated that this phenomenon might be related to bamboo shoot size. Large-sized *Ph. edulis* shoots can meet the morphological requirements for rapid bamboo shoot growth, while accelerated fiber lignification in the BSD improves its mechanical support.

Furthermore, taste quality had noticeable interspecific and positional effects, but the response of BSD and BSF significantly differed among indicators. Phenolic acids (i.e., tannin and oxalic acid) are strongly photosensitive and can subsequently be affected by sunlight [39,40]. As an external bamboo shoot tissue, BSF is more likely to be exposed to sunlight and thus produce phenolic acids. On the other hand, BSF has a protective effect on BSD, which is more susceptible to environmental stress, disease, pests, etc., and is more likely to produce oxalic acids and tannins to improve osmoregulation [15]. Relative to BSD, however, BSF palatability will reduce. Oxalic acid and tannin exhibited a significant component related response (*p* < 0.001), which had no significant effect on soluble sugar and the sugar:acid ratio, suggesting that phenolic acids are the main determining factor of BSD and BSF palatability. Additionally, total acid, oxalic acid, and tannin content in the shoots of both bamboo species increased with shoot height. This finding was consistent with that of other studies on *Dendrocalamus latiflorus* [41], *D. hamiltonii* [42], and *Bambusa oldhamii* shoots [43]. Moreover, soluble sugar and sugar:acid ratios were highest in the shoot base segments, indicating that the palatability of the shoot base was better. Additionally, these two *Ph. violascens* shoot components were lower in oxalic acid and tannin content and higher in sugar content, indicative of better palatability relative to *Ph. edulis*.

Various amino acid compounds and proportions were responsive to different bamboo shoot components while exhibiting apparent positional and genetic effects, which closely correlated to bamboo shoot palatability [3,44]. BSF total amino acid, essential amino acid, bitter amino acid, umami amino acid, and sweet amino acid flavor compounds in both bamboo species were higher than that of BSD. This may be because divisions in the intercalary meristem grow by segments after emerging from the soil, and internode elongation and thickening require amino acid accumulation to provide the necessary nutrients [15,45]. At the same time, amino acids can also form organic acids, sugars, and other products through the tricarboxylic acid (TCA) cycle, glycolysis, and other metabolic pathways [46], potentially leading to differences in BSF and BSD phenolic acid and sugar content. This also indicates that the interactions among the various nutrient and flavor compounds in bamboo shoots, impact shoot quality. Total amino acid and the four key amino acids (i.e., essential amino acid, bitter amino acid, umami amino acid, and sweet amino acid flavor compounds) in the upper and middle BSD and BSF segments of both bamboo species were high, but consistent with compositional nutrient changes, which can also enhance nutrient accumulation (i.e., protein, carbohydrates, etc.) [18,47] and promote the rapid elongation of upper internodes. Additionally, the content and proportion of bitter amino acid flavor compounds in *Ph. edulis* was higher, dramatically affecting shoot taste quality. In contrast, proportionally, umami and sweet amino acid flavor compounds in *Ph. violascens* bamboo shoots were high, improving palatability.

## 5. Conclusions

This study found significant shoot nutrition and flavor differences between the bamboo shoot flesh (BSF) and shoot diaphragm (BSD), along with pronounced positional and interspecific effects. Results showed that BSF nutrient and amino acid content in both bamboo species was higher than that of BSD. In contrast, BSD had comparatively lower total acid, oxalic acid, and tannin content and higher soluble sugar and sugar:acid ratios. Generally, *Ph. violascens* shoot BSD, and BSF palatability was better, with lower acidity, higher sugar, and proportionally higher sweet and umami amnio acid flavor compounds relative to *Ph. edulis*, making it the more palatable choice as a fresh bamboo product. Furthermore, *Ph. edulis* was richer in nutrients and amino acids in its upper BSF segment (i.e., poorer palatability). In contrast, the sugar content in the base segment of the BSF was higher (i.e., better palatability) relative to the BSD of *Ph. violascens*, which can be further classified and utilized.

## Figures and Tables

**Figure 1 foods-12-01180-f001:**
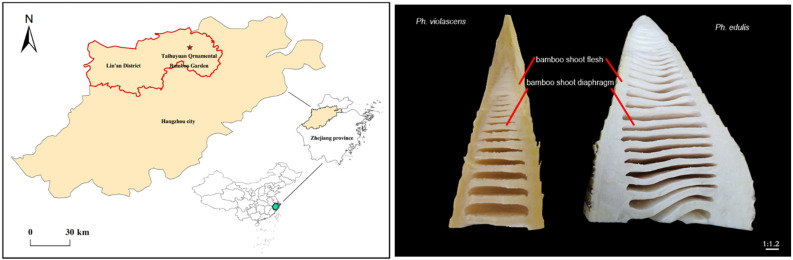
Geographic location of the Taihuyuan Ornamental Bamboo Garden in China (**left**) and a photograph showing the bamboo shoot diaphragm and flesh of *Ph. edulis* and *Ph. violascens* (**right**). The pentagram means the location of study site in the Taihuyuan Ornamental Bamboo Garden, Lin’an City, Zhejiang Province, China.

**Figure 2 foods-12-01180-f002:**
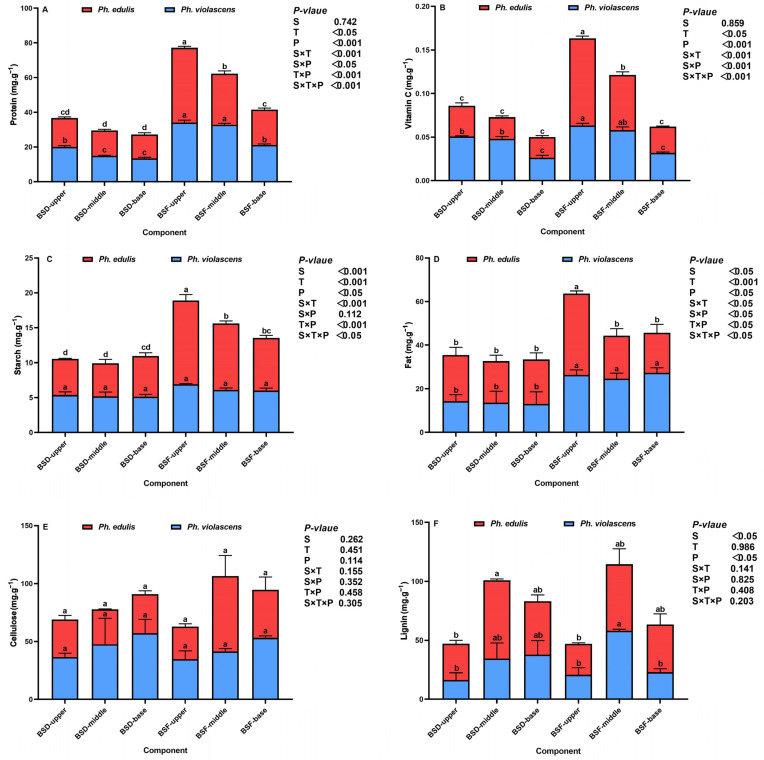
Changes in the nutritional status of bamboo shoot diaphragm (BSD) and flesh (BSF) between *Ph. edulis* and *Ph. violascens*. (**A**) protein, (**B**) starch, (**C**) fat, (**D**) vitamin C, (**E**) cellulose, and (**F**) lignin. BSD and BSF refer to bamboo shoot diaphragm and flesh, respectively. Vertical bars are means ± SD of the four replicates. Different lowercase letters above the bars in each graph represent significant differences (*p* < 0.05) between the bamboo shoot segments. *p* values are only shown for the main and interactive effects of species (S), component (C), and position (P), respectively, of each indicator.

**Figure 3 foods-12-01180-f003:**
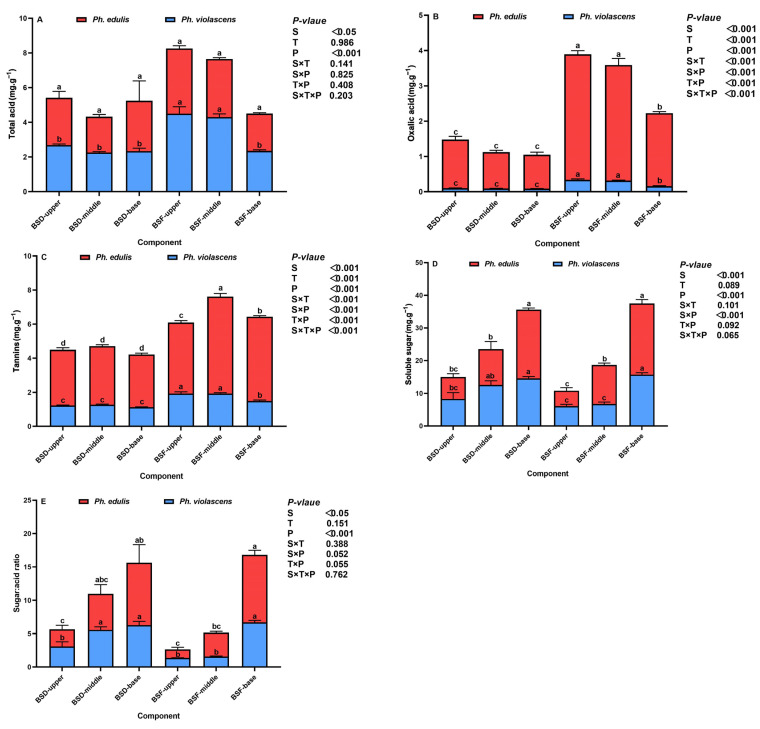
Changes in the nutritional status of bamboo shoot diaphragm (BSD) and flesh (BSF) in *Ph. edulis* and *Ph. violascens*. (**A**) total acid, (**B**) oxalic acid, (**C**) tannins, (**D**) soluble sugar, and (**E**) the sugar:acid ratio. BSD and BSF refer to bamboo shoot diaphragm and flesh, respectively. Vertical bars are means ± SD of the four replicates. Different lowercase letters above the bars in each graph represent significant differences (*p* < 0.05) between the bamboo shoot segments. *p* values are only shown for the main and interactive effects of species (S), component (C), and position (P), respectively, for each indicator.

**Figure 4 foods-12-01180-f004:**
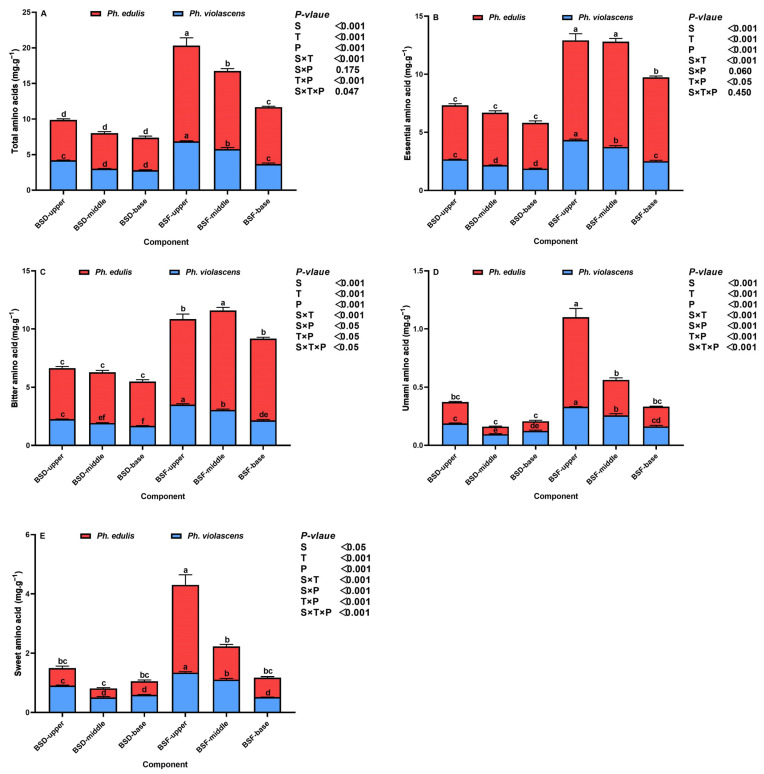
Changes in the amino acid composition of bamboo shoot diaphragm (BSD) and flesh (BSF) of *Ph. edulis* and *Ph. violascens*. (**A**) Total amino acid, (**B**) essential amino acid, (**C**) bitter amino acid, (**D**) umami amino acid, and (**E**) sweet amino acid flavor compounds. BSD and BSF refer to bamboo shoot diaphragm and flesh, respectively. Vertical bars are means ± SD of the four replicates. Different lowercase letters above the bars in each graph represent significant differences at *p* < 0.05 between the different bamboo shoot components. *p* values are only shown for the main and interactive effects of species (S), components (C), and positions (P), respectively, of each indicator.

**Figure 5 foods-12-01180-f005:**
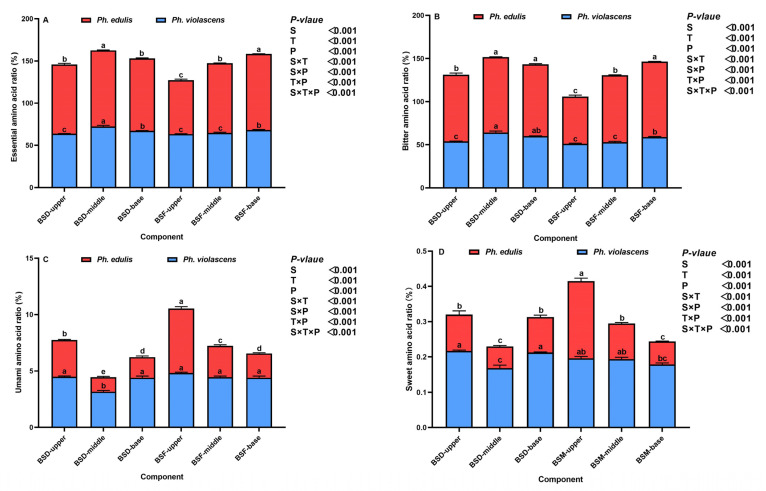
Proportional changes in amino acid composition in bamboo shoot diaphragm (BSD) and flesh (BSF) of *Ph. edulis* and *Ph. violascens*. (**A**) Essential amino acid ratio, (**B**) bitter amino acid ratio, (**C**) umami amino acid ratio, and (**D**) sweet amino acid ratio. BSD and BSF refer to bamboo shoot diaphragm and flesh, respectively. Vertical bars are means ± SD of the four replicates. Different lowercase letters above the bars in each graph represent significant differences at *p* < 0.05 between the bamboo shoot segments. *p* values are only shown for the main and interactive effects of species (S), components (C), and positions (P), respectively, of each indicator.

## Data Availability

The data presented in this study are available in the article.

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
