# Peer review of "Shoot Nutrition and Flavor Variation in Two Phyllostachys Species: Does the Quality of Edible Bamboo Shoot Diaphragm and Flesh Differ?"

_foods, 2023, doi:10.3390/foods12061180_

Round 1
Reviewer 1 Report
The manuscript “Shoot nutrition and flavor variation in two Phyllostachys species: Does the quality of edible bamboo shoot diaphragm and meat differ?” au nutritional differences in two species of Phyllostachys. Here are the considerations: - Item 2.2: How were the two species identified? It is not clear in the methodology. - Item 2.3: Literature references are missing. - Shouldn't carrying out that flavor analysis includes sensory analysis? - The article makes a detailed characterization of the two species. The discussion failed to mention what are the contributions in the practice of characterizing these two bamboo species. - Are the analysis methods used fast, easily accessible and at a cost that favors practical implementation to distinguish between the two species? - Conclusion: “Bamboo shoot quality is affected by genetics, the surrounding environment, local cultivation practices, anthropogenic activities, etc.” This is not a conclusion of the work based on the results found.
Reviewer 2 Report
Manuscript titled, shoot nutrition and flavor variation in two Phyllostachys species: Does the quality of edible bamboo shoot diaphragm and meat differ? Discuss the nutritional and other parameters of bamboo shoot. Title looks interesting but it is misleading the readers. Authors mentioned meat, which is not correct in the case of this study. Although the manuscript is well written, the quality of the manuscript is poor with very weak experimental methods.
Does the quality of edible bamboo shoot diaphragm and meat differ? How can authors make this statement? Are they compared with real meat samples? Based on this study, I have not seen any such studies. In this case, authors must review the title.
We cannot use the word meat. How can the author confirm this as a shoot meat? Authors must revise the word meat. MEAT is dedicated to only animal sources.
If the authors have evidence to show, they can use the word meat. Based on my eating experience of bamboo shoots, I never felt that shoots are like meat.
Abstract
The background of the study should be clearer. Authors must revise this
There is no information on sheet meat. Authors should remove the word meat from the entire manuscript. It is not appropriate to use the word sheet meat
Meat of bamboo? It is not appropriate to say meat. Authors must revise this word
Line 98: see can be removed from see figure 1. Please revoke see throughout the manuscript
The material that comes from plant sources is not meat. Do you have evidence to show it is meat?
Line 128: citation is not in correct format.
Section 2.3. Methodology should be explained in detail.
Hplc method and other methods should be provided with complete details.
As such, the methodology is very weak. I suggest providing full information of each method along with the following literature citation.
Figure 2 quality must be improved
Using the word meat is not appropriate for this study. Authors must revise it
Figure 3 quality must be improved
All figures quality must be improved
The presentation of the results is fine but discussion must be improved and must compare with more recent literature.
Conclusions must be revised to reflect the research findings.
Ref are not according to the journal format.
Round 2
Reviewer 2 Report
Authors are now addressed the suggestions made by me. However, the following point is not addressed.
Abstract
The background of the study should be clearer. Authors must revise this
I suggest authors to provide clear background in abstract. Fir two lines (lines 11 and 12-13) are not connecting with study objectives.
